# Capture and visualization of live *Mycobacterium tuberculosis* bacilli from tuberculosis patient bioaerosols

Ryan Dinkele[1,2☯], Sophia Gessner[1,2☯], Andrea McKerry[3], Bryan Leonard[3], Ronnett Seldon[3], Anastasia S. Koch[1,2], Carl Morrow[2,3], Melitta Gqada[3], Mireille Kamariza[4], Carolyn R. Bertozzi[5,6], Brian Smith[7], Courtney McLoud[7], Andrew Kamholz[7], Wayne Bryden[8], Charles Call[8], Gilla Kaplan[9], Valerie Mizrahi[1,2,10], Robin Wood[2,3]*, Digby F. Warner[1,2,10]*

1 SAMRC/NHLS/UCT Molecular Mycobacteriology Research Unit & DST/NRF Centre of Excellence for Biomedical TB Research, Department of Pathology, Faculty of Health Sciences, University of Cape Town, Cape Town, South Africa, 2 Institute of Infectious Disease and Molecular Medicine, Faculty of Health Sciences, University of Cape Town, Cape Town, South Africa, 3 Desmond Tutu HIV Centre, University of Cape Town, Cape Town, South Africa, 4 Department of Biology, Stanford University, Stanford, California, United States of America, 5 Department of Chemistry, Stanford University, Stanford, California, United States of America, 6 Howard Hughes Medical Institute, Stanford University, Stanford, California, United States of America, 7 Edge Embossing, Boston, Massachusetts, United States of America, 8 Zeteo Tech, Sykesville, Maryland, United States of America, 9 Department of Medicine, University of Cape Town, Cape Town, South Africa, 10 Wellcome Centre for Infectious Diseases Research in Africa, University of Cape Town, Cape Town, South Africa

☯ These authors contributed equally to this work.
* robin.wood@hiv-research.org.za (RW); digby.warner@uct.ac.za (DFW)

**Data Availability Statement:** All relevant data are within the manuscript and its supporting information files.

## Abstract

Interrupting transmission is an attractive anti-tuberculosis (TB) strategy but it remains under-explored owing to our poor understanding of the events surrounding transfer of *Mycobacterium tuberculosis* (*Mtb*) between hosts. Determining when live, infectious *Mtb* bacilli are released and by whom has proven especially challenging. Consequently, transmission chains are inferred only retrospectively, when new cases are diagnosed. This process, which relies on molecular analyses of *Mtb* isolates for epidemiological fingerprinting, is confounded by the prolonged infectious period of TB and the potential for transmission from transient exposures. We developed a Respiratory Aerosol Sampling Chamber (RASC) equipped with high-efficiency filtration and sampling technologies for liquid-capture of all particulate matter (including *Mtb*) released during respiration and non-induced cough. Combining the mycobacterial cell wall probe, DMN-trehalose, with fluorescence microscopy of RASC-captured bioaerosols, we detected and quantified putative live *Mtb* bacilli in bioaerosol samples arrayed in nanowell devices. The RASC enabled non-invasive capture and isolation of viable *Mtb* from bioaerosol within 24 hours of collection. A median 14 live *Mtb* bacilli (range 0–36) were isolated in single-cell format from 90% of confirmed TB patients following 60 minutes bioaerosol sampling. This represented a significant increase over previous estimates of transmission potential, implying that many more organisms might be released daily than commonly assumed. Moreover, variations in DMN-trehalose incorporation profiles suggested metabolic heterogeneity in aerosolized *Mtb*. Finally, preliminary analyses indicated

**Funding:** The authors acknowledge the financial support of the South African Medical Research Council (www.samrc.ac.za) with funds from National Treasury under its Economic Competitiveness and Support Package (MRC-RFA-UFSP-01-2013/CCAMP, R.W.), for Extramural Unit funding (to V.M.), and via the Strategic Health Innovations Partnerships Unit (www.samrc.ac.za/innovation/strategic-health-innovation-partnerships) (to D.F.W and V.M). We are grateful for funding from the Bill and Melinda Gates Foundation (www.gatesfoundation.org; OPP1116641, R.W.), the Research Council of Norway (www.forskningsradet.no; R&D Project 261669 "Reversing antimicrobial resistance", D.F. W.), the Eunice Kennedy Shriver National Institute of Child Health and Human Development (www.nichd.nih.gov; U01HD085531, D.F.W.), and the US National Institute of Allergy and Infectious Diseases (www.niaid.nih.gov; R01AI147347, R.W.). We also acknowledge the Howard Hughes Medical Institute (www.hhmi.org) for a Senior International Research Scholars grant (V.M.), Stanford University's Diversifying Academia, Recruiting Excellence Fellowship (www.stanford.edu), and the NIH Predoctoral Fellowship F31AI129359 (M.K), the Bill and Melinda Gates Foundation (OPP115061) and NIH (AI051622) grants (to C.R. B.), and the Carnegie Corporation of New York (www.carnegie.org) via sub-award from the University of Cape Town (www.uct.ac.za) (to A.K). The funders had no role in study design, data collection and analysis, decision to publish, or preparation of the manuscript.

**Competing interests:** The authors have declared that no competing interests exist.

the capacity for serial image capture and analysis of nanowell-arrayed bacilli for periods extending into weeks. These observations support the application of this technology to long-standing questions in TB transmission including the propensity for asymptomatic transmission, the impact of TB treatment on *Mtb* bioaerosol release, and the physiological state of aerosolized bacilli.

## Author summary

*Mycobacterium tuberculosis* (*Mtb*), the cause of tuberculosis (TB), must drive successive cycles of transmission and infection to retain a foothold in its obligate human host. Although critical for *Mtb* survival, the mechanisms enabling successful transmission have largely evaded research owing to the difficulties inherent in identifying when bacilli are released and by whom. With the available tools, fewer than one-third of new *Mtb* infections can be confidently linked to known TB cases, a deficiency reflecting the confounding effects of the prolonged infectious period of TB and the potential for transmission from transient exposures. Here, we describe the deployment of the Respiratory Aerosol Sampling Chamber (RASC), a personal clean room equipped for high-efficiency capture of bioaerosols, to isolate live *Mtb* bacilli released in infectious aerosols. Applying a fluorescent viability probe and microscopic imaging, we demonstrate the detection of live *Mtb* with single-cell resolution in complex bioaerosol samples from a high proportion of TB cases. Moreover, by exploiting compartmentalization of bacilli within a nanowell collection device, we establish the capacity for long-term maintenance of bacillary viability for serial imaging. Our observations support the utility of the RASC to better understand and ultimately interrupt *Mtb* transmission.

## Introduction

*Mycobacterium tuberculosis* (*Mtb*), the causative agent of tuberculosis (TB), is the leading infectious killer globally, claiming ~1.4 million lives annually [1]. TB control is heavily predicated on treatment of active disease. However, delayed and missed diagnoses, and the six-month duration of standard chemotherapy, contribute to failure of this approach to control the TB epidemic [2]. The increasing prevalence of drug-resistant TB, compounded by evidence of the transmission of multi-drug resistant (MDR) *Mtb* strains, further undermines this approach [3]. Together, these factors have focused attention on blocking *Mtb* transmission as a critical area for novel interventions [4]. In turn, this has highlighted significant gaps in our knowledge of *Mtb* transmission: fewer than 30% of new *Mtb* infections can be linked to a known TB case, suggesting the existence of unrecognized transmitters in TB endemic communities [5].

Reconstruction of transmission chains has traditionally required genetic fingerprinting of *Mtb* strains isolated from active TB cases and their diseased contacts. Despite advances in molecular epidemiology, this has proven enormously challenging even in low-incidence settings [6]. Consequently, the host and mycobacterial factors which ensure successful transmission remain obscure [7]. Since the interval between the time of infection and diagnosed disease varies, analyses are necessarily retrospective, making intervention impossible [8]. Moreover, targeting surveillance to active disease means that the potential contribution of asymptomatic transmitters is overlooked [9]. The question of whether asymptomatic individuals are able to transmit *Mtb* is especially relevant for TB control: to date, community-based

exposure studies have focused predominantly on household contacts which account for fewer than 20% of infections in high TB-burden settings [10].

Direct study of aerosolized *Mtb* is equally complicated: issues such as timing of sampling, the small numbers of bacilli released in exhaled air and sputum, and the presence of environmental and patient-derived contaminating microorganisms and particulate matter impose profound technical and analytical challenges [11, 12]. Enumeration of viable aerosolized *Mtb* via microbiological culture is complicated by the semi-quantitative nature of "time to positivity" in liquid culture and the slow formation of colony forming units (CFU) on solid media (four to eight weeks for colonies to become visible) [13]. Additional complications arise from the presence in clinical *Mtb* samples of "differentially detectable" organisms [12]–which means that microbiological culture often underestimates the true size of the viable bacillary population, and the temporal–and, consequently, genetic [14] and physiological–separation of the (single) transmitted bacillus from the micro-population (~$10^6$ cells) contained in a visible *Mtb* colony on a plate. Decontamination, too, undermines assessments of bacillary load and physiological state. Where excessive, it depletes the number and viability of *Mtb* bacilli in the sample [15]; where inadequate, it risks overgrowth by contaminants, obscuring the signal.

Molecular methods have enabled detection of *Mtb* DNA in bioaerosols [16], however they do not distinguish live from dead organisms, and even protocols which target RNA [17] require extraction of intracellular nucleic acid, obviating the potential to investigate the physiological and metabolic state(s) of aerosolized bacilli. The method of bacillary capture is also important: approaches based on cough assume symptomatic spread, ignoring the possibility of *Mtb* transmission during normal respiratory activity and, therefore, potentially not capturing other natural transmission events [18], especially those from sub-clinical infections [19]. Facemask and equivalent sampling methodologies either render live-cell analyses impossible or *in vitro* propagation (via microbiological culture) unavoidable [20]. In addition, many methods are unable to determine whether bacilli derive from small (buoyant) or large aerosol droplets. This is a critical flaw given the likelihood that size determines aerosol longevity and ability to access lung alveoli, key elements of infectiousness [21].

To address these challenges, we sought to develop a method for culture-independent detection, quantification, and visualization of live bacilli in bioaerosols captured using the Respiratory Aerosol Sampling Chamber (RASC), a small personal clean-room enabling capture of particulate material released by an individual patient during normal respiratory activity, including natural (non-induced) cough. In previous work, we demonstrated the potential for liquid capture of aerosolized *Mtb* in the RASC, eliminating the dependency on solid culture-based techniques for bacillary detection [11, 12]. This was a key innovation since it opened the possibility for detection, isolation, and manipulation of live bacilli for downstream phenotypic and genomic studies. Here, we advance this technology in a new cohort of recently diagnosed TB patients, incorporating methodologies for the specific labelling and enumeration of low numbers of live *Mtb* bacilli from aerosol samples in a format which enables detection via live-cell fluorescence microscopy of single bacilli arrayed in a nanowell device. Moreover, by assigning localization co-ordinates to individual fluorescent organisms in the nanowells, we demonstrate the capacity to extend microscopic analyses to live cells for prolonged durations under incubation.

## Results

### A custom-built nanowell device for microscopic analyses

The fluorescent trehalose analogue, 4-*N*, *N*-dimethylsamino-1,8-napthalimide-trehalose (DMN-trehalose), enables rapid labelling and microscopic detection of *Mtb* in sputum or

liquid medium [22]. Since active membrane biosynthesis is necessary for DMN-trehalose incorporation, this probe possesses the advantage of labelling live, metabolically active organisms only. Moreover, the solvatochromic properties of the DMN fluorophore mean that the signal is enhanced following incorporation into the mycobacterial cell envelope, limiting background noise and thus circumventing the need for multiple wash steps–a critical consideration when aiming to detect all *Mtb* in a potentially paucibacillary (<100 bacilli/ml) bioaerosol sample. To enhance our capacity for isolation and DMN-trehalose-enabled microscopic detection of single *Mtb* cells from bioaerosol, we developed a custom-designed device comprising arrays of 50 x 50 μm nanowells (Fig 1A and 1B). Physical separation of samples across thousands of individual nanowell compartments was considered beneficial in ensuring that (i) any contaminating particulate matter would be distributed across the nanowells and (ii) all viable organic material would be isolated in discrete nanowell chambers, reducing the likelihood that faster-growing non-*Mtb* organisms ("contaminants" for TB diagnostic purposes, but natural components of aerosol microbiomes) might overwhelm the device following overnight labelling.

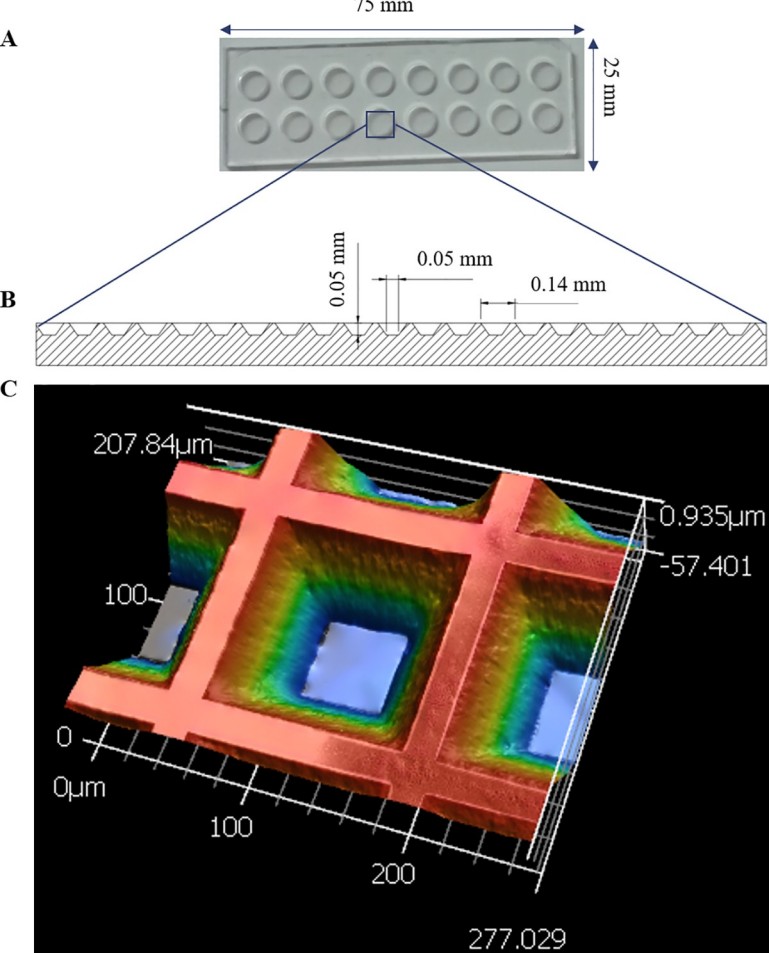

**Fig 1. Design and fabrication of nanowell-arrayed microscope slides for the compartmentalization and visualization of TB bioaerosols.** (A) Photograph and (B) schematic of a nanowell-arrayed microscope slide. (C) 3D scan of a 207.840 x 277.029 μm section of the slide. Each device (25 mm x 75 mm) consists of two rows of eight round microwells machined from cast acrylic. The microwells are 6 mm in diameter and 2 mm deep. The nanowell film, which is bonded to the superstructure with UV-curing adhesive, is made from embossed COC film. The nanowells have side-wall angles of 35° and are 50 μm deep. The distance through the bottom of each well to the back of the film is ~170 μm, equivalent to a number 1.5 coverslip.

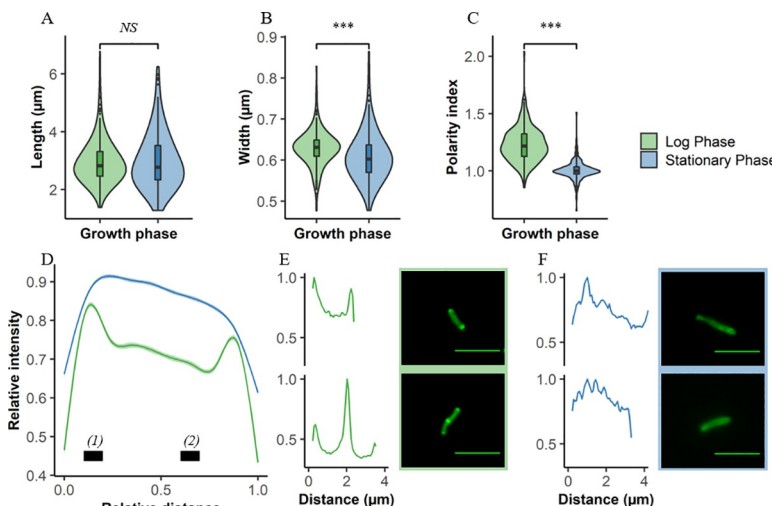

**Fig 2. Differentiation of growth states in *Mtb* using DMN-trehalose and cell morphology.** Comparisons between log (green) and stationary (blue) phase *Mtb* according to (A) cell length, (B) width, and (C) polarity index. (D) Average DMN-trehalose profile for log and stationary phase bacilli, with single-cell examples in both (E) log and (F) stationary phase. Polarity index for each cell was calculated as the median fluorescence intensity at region *(1)* divided by the median fluorescence intensity at region *(2)* in panel (D). Wilcoxon signed-rank test performed, p < 0.001 = ***, NS = not significant.

## Towards an identikit of DMN-trehalose-positive *Mtb* bacilli

In developing a framework (or "identikit") for the assignment of "putative *Mtb* bacillus" to individual fluorescent structures/ microparticles detected in the nanowell device, we processed exponentially growing and aged cultures of the laboratory strain, *Mtb* H37Rv, via the DMN-trehalose labelling protocol to assess the potential for morphological variation as a function of metabolic and replicative status. The median length of *Mtb* H37Rv bacilli (Fig 2A) did not change between exponential growth (2.83 μm, IQR = 0.85 μm) and early stationary phase (2.77 μm, IQR = 1.19 μm). In contrast, there was a small but statistically significant difference in average cell width (Fig 2B), with bacilli entering early stationary phase (0.60 μm, IQR = 0.07 μm) very slightly thinner than exponentially growing organisms (0.63, IQR = 0.04 μm). However, the average differences observed in mycobacterial cell width were unable to differentiate log from stationary phase bacilli owing to the largely overlapping distributions. This observation, plus the knowledge that bacillary length is commonly reported as a feature of morphological heterogeneity of clinical *Mtb* isolates [23], supported the adoption of width (~0.47 μm to ~0.86 μm) and DMN-trehalose positivity as primary markers for the identification of "putative *Mtb*" in bioaerosol samples.

The metabolic state of aerosolized *Mtb* remains unknown. Therefore, we investigated the utility of cytological profiling via DMN-trehalose staining to differentiate bacilli broadly as either slow or fast growers based on their growth phase. Substantial differences in DMN-trehalose uptake and distribution along the cell length were observed between log and stationary phase bacilli (Fig 2C–2F). The polarity index, a simple metric of the relative brightness of the pole compared to the mid cell, was greater in log phase (1.21, IQR = 0.242) compared to stationary phase cells (0.948, IQR = 0.112) (Fig 2C). As such, cells identified as putative *Mtb* based on cell width and DMN-trehalose positivity could be characterized broadly into at least two categories based on DMN-trehalose staining.

DMN-trehalose is not specific for *Mtb*: all bacteria within the Actinomycetales which encode homologs of the mycobacterial antigen 85 complex possess the capacity to incorporate

**Table 1. Summary of samples investigated.**

| | | Total | Positive (%) | Median count[†] |
|---|---|---|---|---|
| Sample type | TB+ patient | 31 | 28 (90.3) | 14.0 |
| | Empty RASC | 26 | 14 (53.8) | 1.5 |
| | Total | 57 | | |

† Median number of putative *Mtb* bacilli detected in the sample

the fluorescent label. Among these, Corynebacteria are a common constituent of the oral microbiome and were previously identified in TB sputum samples [22]. Therefore, to expand our database of potential DMN-trehalose-positive (DMN-tre+) organisms, we included the opportunistic pathogen, *Corynebacterium striatum* [24], in the *in vitro* analyses. Following DMN-trehalose labelling, *C. striatum* was readily distinguishable from mycobacteria, yielding a distinct cytological profile characterized by an ovoid cell shape with near-uniform fluorescence signal throughout the cell membrane and septa (S1A Fig, panel i).

### Microscopic identification and characterization of DMN-tre+ *Mtb* in TB bioaerosols

Thirty-one individuals with GeneXpert-positive, drug-susceptible TB were recruited into a pilot study (Table 1) for the capture and single-cell detection of live, *Mtb* bacilli in patient bioaerosols (Fig 3). Putative DMN-tre+ *Mtb* was identified based on cell width and DMN-trehalose positivity in 90% (28/31) of the TB patient bioaerosols (Table 1).

The median count in the captured air from TB-positive patients was 14 (range 0–36) versus 1.5 (range 0–10) in the air analyzed from the empty RASC booths suggesting low-level carry over/contamination of the booth between tested individuals (Fig 4A). When comparing the basic morphological characteristics of aerosolized *Mtb* to those observed in log-phase, it was apparent that they were significantly shorter (Fig 4B) but with the same width on average (Fig 4C). However, a greater degree of width variation was observed in clinical samples (Fig 4C), suggesting that application of the selected width criteria (~0.47 μm to ~0.86 μm) to patient aerosols might be conservative.

Further investigation into the profiles of DMN-trehalose incorporation led to the observation of three distinct labelling patterns (Fig 4D–4F), namely: polar labelling (sample TRDS182,

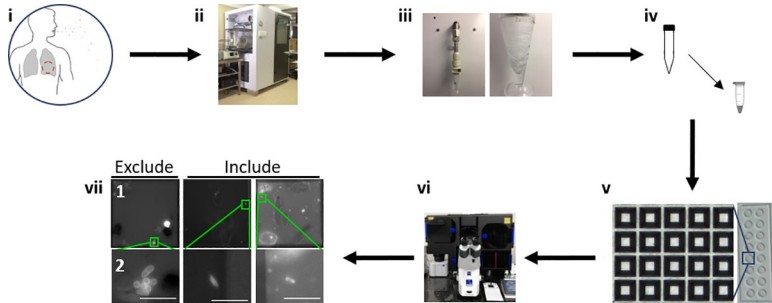

**Fig 3. Workflow from participant recruitment to image analysis.** (i) Recruitment of TB GeneXpert+ patients. (ii) One hour of bioaerosol production during tidal breathing and non-induced cough within the Respiratory Aerosol Sampling Chamber (RASC). (iii) Liquid capture of patient bioaerosol via Bertin Coriolis μ Biological Air Sampler. (iv) Bioaerosol concentration and staining with 100 μM DMN-trehalose during overnight (~16 hours) incubation at 37 °C. (v) Sample arraying within the nanowell device. (vi) Manual sample scanning and bacilli enumeration. (vii) Nanowell imaging (row 2 represents a zoomed in section from row 1). Columns represent 3 different patients. Bacilli not matching inclusion criteria are excluded from subsequent analysis. Scale bar, 5 μm.

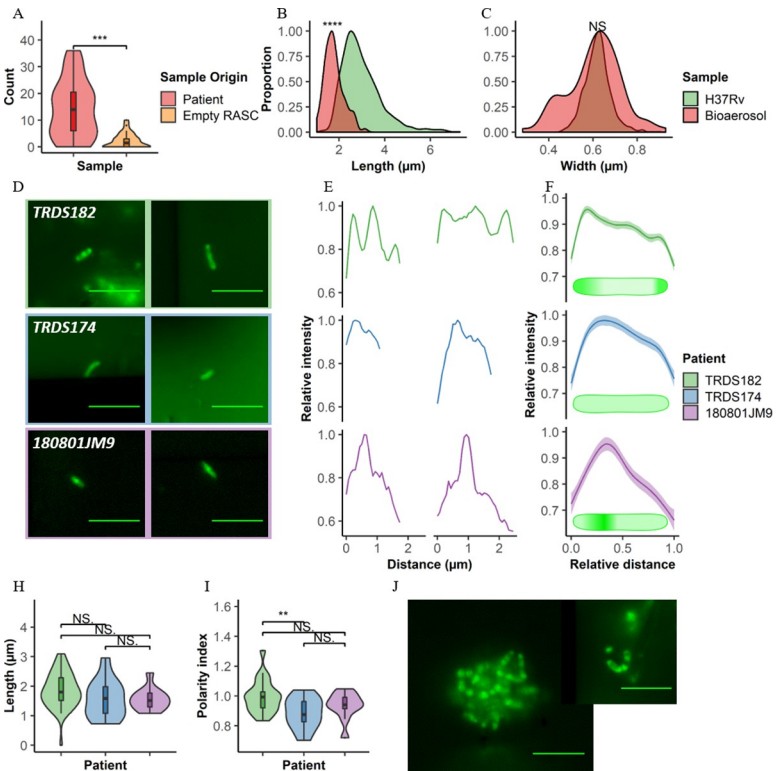

**Fig 4. Detection and characterization of putative *Mtb* within bioaerosols of confirmed TB patients.** (A) Plot comparing the number of putative *Mtb* detected within TB[+] participants (red, n = 31) and empty RASC controls (orange, n = 27). Comparing distributions of (B) cell lengths and (C) cell widths in putative *Mtb* bacilli detected within bioaerosols of TB patients (red) to *Mtb* H37Rv cultured within the lab (green). Representative (D) images and (E) plots of the three distinct, exemplar cytological profiles from three patients in which putative *Mtb* were detected. (F) Average plots and idealised drawings indicating the different staining patterns of all bacilli detected in these three patients. (H) Polarity index and (I) length of the bacilli detected within these patients. (J) Representative images of clumps of putative *Mtb* detected within bioaerosol samples (TRDS182). Scale bar = 5 μm. Wilcoxon Rank-Sum test performed, $p < 0.01 = $ **, $< 0·001 = $ ***, $p < 0,0001 = $ ****, NS = not significant.

top row), diffuse labelling (sample TRDS174, middle row), and a patchy labelling pattern (sample 180801JM9, bottom row). Both polar and diffuse labelling have been previously observed in log and stationary phase bacilli, respectively (Fig 2D). Interestingly, this patchy labelling pattern wasn't common in our *in vitro* experiments. Like the *in vitro* cultured organisms, no significant differences in cell length were observed for the different labeling patterns in the bioaerosol samples (Fig 4H). However, more prominent differences were seen in labelling pattern of DMN-trehalose (Fig 4I), highlighting the potential utility of the trehalose probe in indicating underlying metabolic states of aerosolized bacilli. Additional surprising results were the observation of clumps and small clusters of organisms; however, these were not common and were observed in only a fraction (2/31) of patients (Fig 4J).

Numerous organisms were detected with features closely matching those observed for laboratory-grown *C. striatum* (S1A Fig, panel ii). We also observed multiple DMN-tre[+] organisms of possible bacterial and/ or fungal origin–despite the expectation that the ability to metabolize the fluorescent trehalose analogue should be limited to the Actinomycetales (S1B Fig). Utilizing the morphologic exclusion criteria developed above, all of these were eliminated from "putative *Mtb*" classification despite showing a DMN-tre[+] phenotype (S1B Fig).

## Serial imaging of captured, DMN-tre⁺ *Mtb* bacilli in the arrayed nanowells

A key motivation informing the development of the RASC platform was the need to capture live, aerosol-derived *Mtb* for analysis and propagation as part of a larger research program in TB transmission and *Mtb* aerobiology. This requires the capacity not only to isolate and detect individual bacilli, but also to maintain the viability of organisms in a format amenable to extended analysis and cultivation. To investigate the suitability of the nanowell array for this purpose, we prepared a small subset of bioaerosol samples for extended incubation *in vitro*. The samples were processed according to the standard DMN-trehalose staining protocol but, after the final wash step, were resuspended in fresh Middlebrook 7H9 culture medium before arraying on the nanowell slide and incubating at 37 °C without shaking. Following initial identification of putative DMN-tre⁺ *Mtb* bacilli (Day 0), images were captured every 24 h for the first week (excluding weekends) and again on Day 14 post isolation using the x-y coordinates of the specific nanowell to enable re-location of the same organisms for serial imaging (Fig 5).

Over the two-week incubation period, no changes in length were observed for any of the cells, whether classified as "Putative *Mtb*" or "Other". Similarly, the fluorescence intensities over background for individual cells displayed only minor increases for some of the "Other" organisms. It's possible that the incubation time was too short to allow for adaptation of the bioaerosol-derived bacilli to the culture medium and/or solid substrate, with any alterations in metabolic or replicative state expected to manifest in altered bacillary morphology and/or DMN-trehalose profile; therefore, ongoing work includes exploring different culture media

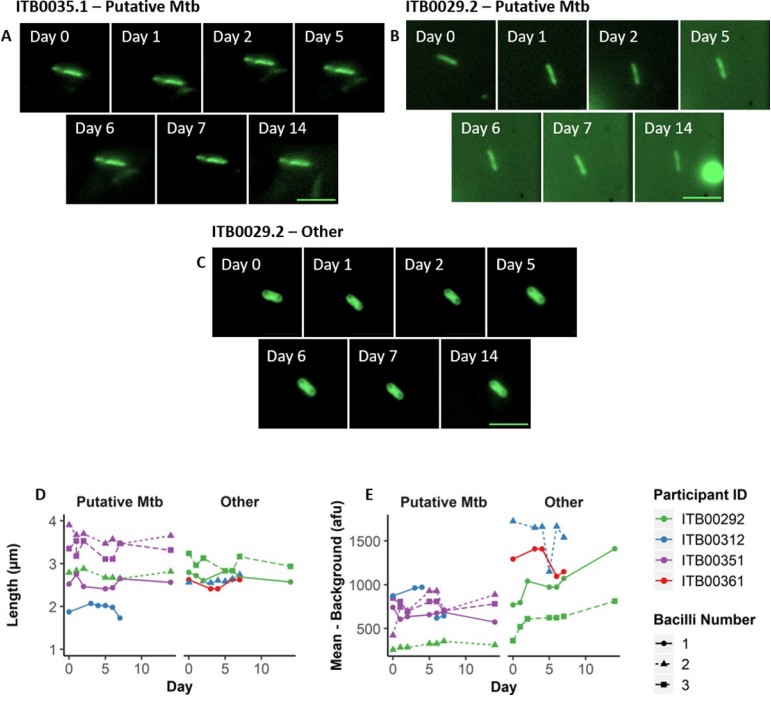

**Fig 5. Serial imaging of organisms captured directly from patient bioaerosol for up to two weeks.** Single bacilli identified from 4 separate patients were serially imaged daily for the first 7 days (except weekends) and weekly thereafter. Up to three bacilli were tracked per patient (bacilli number–represented by shape and dashed lines) in four patients (represented by colour) and identified as either "Putative *Mtb*" or "Other". (A—C) Representative bacilli from two separate patients imaged on days 0, 1, 2, 5, 6, 7, and 14. (A) and (B) represent putative *Mtb*, whereas (C) represents other organisms with a low probability of being *Mtb* based on the applied inclusion criteria. Summary of bacterial changes in (D) length and (E) mean fluorescence intensity minus the average background intensity. Scale bar = 5µm.

and extending the analysis from weeks to months. We were nevertheless encouraged by the ability to obtain serial images of the same organisms, supporting the utility of the nanowell array in enabling re-imaging of DMN-tre[+] bacilli identified immediately post capture and assigned a unique "address" to allow for re-location.

## Discussion

Fluorescence microscopy of bioaerosol samples enabled the detection of putative live *Mtb* in 90% of GeneXpert-confirmed TB patients. While absolute bacillary numbers in exhaled air samples were low (the maximum DMN-tre[+] count was 36), it must be remembered that these samples were collected from only 60 minutes in the RASC and without requirement for a specific respiratory maneuver or induced cough. Consequently, simple extrapolation of the 14 bacilli median would suggest the release of around ~336 viable bacilli per day. This number is consistent with recent estimates based on face-mask sampling [7]. However, direct comparison is difficult owing to the use in that study of quantitative PCR targeting of the IS*6110* locus (present in *Mtb* genomes in variable copy number) with an acknowledged detection limit of ~33 CFU from the gelatin filters. Of note, release of several hundred metabolically active bacilli (inferred from DMN-trehalose incorporation) implies significant infection potential; moreover, the observation here of putative *Mtb* "clumps" resonates with *in vitro* evidence demonstrating the enhanced capacity of *Mtb* aggregates to subvert macrophage antimicrobial defenses [25].

The physiological state of aerosolized organisms remains uncertain owing to the previous unavailability of tools to investigate this stage of the TB disease cycle. Although DMN-trehalose positivity alone implies metabolic activity (bacilli must be viable to incorporate the trehalose analog into the mycomembrane), it is not known if these organisms are replicatively active, nor if aerosolized bacilli contain defining alterations in cell envelope [26] or other macromolecular compositions and/or inclusions [27]. We determined the sensitivity of DMN-trehalose labelling to metabolic state as a function of log *versus* stationary phase growth, recording significant differences that could be used to distinguish these states. Similar profiles were observed in organisms from patient bioaerosols, however the implications of these observations remain uncertain and will require further research, potentially involving the incorporation of more than one spectrally compatible fluorescent probe [28]. The potential exists to extend this work beyond investigating metabolic state, focusing on drug resistance, for example. Previous work has shown differences in probe incorporation profiles in response to antibiotic treatment [29]. At a quantitative level, imaging organisms from diagnosed TB patients before and after treatment initiation, as well as serially over the course of standard chemotherapy, might therefore offer a more rapid indication of drug efficacy, in effect affording the sensitivity of broncho-alveolar lavage without the invasiveness of that procedure.

Considerable effort was made to minimize the acquisition of "contaminating" debris during RASC sampling–for example, by working in a clean-room and requiring that all patients wore disposable biohazard suits to reduce release of non-respiratory particles. This is because the cleanliness of samples is critical for microscopic visualization: auto-fluorescence is a major confounder in environmental samples and debris can obscure *Mtb* bacilli under microscopic investigation (S2 Fig). The nanowell device, too, was designed to maximize *Mtb* detection by increasing the likelihood of separating debris and bacilli through sample dispersal across thousands of nanowell chambers. It is possible that the composition and origin of the particulate matter will prove invaluable in future in determining the anatomical origin of aerosolized bacilli. In this work, though, raising the signal from putative *Mtb* above background noise was our priority. Our "empty booth" controls–in which sampling was performed in the empty

RASC in the absence of an incumbent individual and was therefore expected to be *Mtb* free–returned 54% (14/26) positivity; however, the difference in median count compared to the TB-positive patients suggested that these organisms were carried over from the cyclone collection system or RASC (Fig 4A). Moreover, improvements to the collection system subsequent to this study have ensured complete sterilization, eliminating carryover as confounder.

Bioaerosols impose numerous additional complications which are not applicable to the current standard clinical TB specimens, such as sputum. For example, in processing our samples, we deliberately omitted a decontamination step (sample loss is too significant a factor when dealing with tens of organisms), risking the potential for proportionally larger numbers of non-*Mtb* DMN-trehalose-positive organisms to be present in the bioaerosol samples than in decontaminated sputum. The amplification of fluorescence consequent on incorporation into the mycomembrane suggests the possible use of DMN-trehalose in detecting *Mtb* within untreated and decontaminated sputum [22]. However, the fact that Ag85-dependent trehalose mycolylation is not unique to mycobacteria [22] complicates the use of this probe on its own to classify aerosol samples as "*Mtb* positive". When the present work was initiated, there was a lack of ancillary *Mycobacterium*-specific probes which could be applied to increase the specificity of the microscopic detection while retaining sample viability (thus excluding standard approaches including auramine and acid-fast staining which involve inactivation steps). More recently, other mycobacterial probes have become available, including (i) the CDG-DNB3 dual fluorescence probe which is activated by the mycobacterial BlaC β-lactamase and retained in the cell wall following covalent modification by the decaprenylphosphoryl-β-D-ribose 2′-epimerase, DprE1 [30]; (ii) the TAMRA-labelled benzothiazinone analogue, JN108, which also targets the mycobacterial membrane protein, DprE1, and according to its developers, is able to differentiate *Corynebacterium* from *Mycobacterium* on the basis of fluorescence localization [31]; and (iii) the Quencher-Trehalose-Fluorophore (QTF) which, like DMN-trehalose is a fluorogenic analogue of the natural substrate of mycobacterial mycolyltransferases [32]. It is too soon to ascertain whether any of these will add value to the analysis; moreover, the overlapping auto-fluorescent signal in many samples urges the development of alternative probes in the red end of the visible spectrum. Current work aims to refine our framework for identification of *Mtb* based on automated detection of fluorophores, utilizing algorithms that capture metabolic and morphological characteristics.

Although we did not detect any clear alterations in morphology or DMN-trehalose incorporation profile, the incubation period was limited to two weeks which might be too short to allow for metabolic and/or replicative adaptation in the captured organisms; ongoing work is extending the duration of incubation, and will also explore the use of other compartmentalized capture devices as alternatives to the COC-embossed nanowell format. At the very least, these results support the ability to capture and maintain single *Mtb* cells in an arrayed format for serial imaging, and hint at the potential to exploit the physical separation of cells into nanowell compartments for clonal propagation of bacilli downstream for genomic and other analyses which require biomass [33].

As for any study describing the development of technologies to investigate a previously occult stage of the infectious disease cycle, the approach detailed here inevitably carries inherent limitations which must be considered when interpreting the data: (i) We have not presented orthogonal data confirming the identity of the "putative *Mtb*" identified microscopically. The presumptive evidence, however, is strong: the bioaerosol samples were obtained from GeneXpert-positive TB patients immediately after diagnosis and before treatment initiation, and our previous work has demonstrated the isolation in the RASC platform of PCR-confirmed *Mtb* colony forming units in bioaerosol samples [11, 12]. Moreover, ongoing work in which two samples collected from the same individual are analysed via DMN-

trehalose probing and either auramine staining or RD9 PCR detection have confirmed 100% positivity correlation. (ii) The bacillary counts and morphological phenotypes presented here are not augmented with clinical metadata (chest radiography scores, HIV status, *etc.*). Our intention in this study was to establish the technological platform for bioaerosol sample capture and analysis–as described in our recently published protocol [34], current and planned work involves the application of the RASC to carefully designed clinical cohorts. (iii) The criteria used to classify DMN-tre$^+$ organisms as "putative *Mtb*" are potentially restrictive, especially given that prevailing assumptions about the size and shape of clinical *Mtb* isolates are heavily influenced by the commonly applied staining methods (almost never supported by confirmatory molecular or microbiological data) as well as knowledge of *Mtb* morphology from growth *in vitro* in defined culture, in some cases in intracellular infection models or under applied stress conditions. There is a strong likelihood that we are failing to detect *Mtb* which do not conform to these criteria and, moreover, our use of fluorescence positivity necessarily excludes organism which might be transiently inactive or quiescent [35]. We are exploring the incorporation of automated image detection software to facilitate machine-driven detection of "interesting" structures following capture of microscopy images for all particulate matter (organic and inorganic) arrayed on the nanowell slides. This development is proposed to address a further limitation of our approach, namely that our method relies on detecting then imaging cells (including microscope focusing) based on DMN-trehalose fluorescence. The noise in some samples means finding an optimal focus can be a challenge, which may artifactually increase or decrease the width (and to some degree the length) of the bacilli we measure.

Our priority now is the deployment of the RASC to identify viable *Mtb* in aerosol collected from potentially infectious subclinical cases [19]. The capacity for non-invasive capture and isolation of viable *Mtb* from bioaerosol within 24 hours of collection also supports the potential utility of the RASC to measure the impact of TB treatment on the viability of *Mtb* bioaerosol release. Finally, while this technology enables the detection of viable bacilli, like all aerosol capture methods it does not provide a measure of infectiousness. Ascertaining which *Mtb* isolates go on to infect new individuals and cause TB disease will require innovative approaches toward "closing the loop"; that is, demonstrating productive infection of a new host following release. Historically, this has been achieved primarily by demonstrating infection of animals [36] (thus satisfying a key criterion of Koch's Postulates); the application of genomic epidemiology in combination with RASC-enabled bacillary capture, perhaps in closed community (or even household) settings, offers a modern alternative. For now, the planned modifications to our platform are primarily informed by the need to enable multi-omic analyses of the single-cell organisms isolated from patient aerosols to generate insights that can be pursued in the human host, the natural target of *Mtb*.

## Materials and methods

### Ethics statement

Ethics approval was obtained from the Human Research Ethics Committee, University of Cape Town (HREC 529/2019). Patients were recruited from primary healthcare facilities in Masiphumelele and Ocean View, peri-urban townships located outside Cape Town, South Africa. Informed consent was obtained from all participants and criteria for inclusion were (i) 18 years or older, (ii) GeneXpert-positive TB, and (iii) no evidence of drug resistant TB. All participants were recruited prior to initiation of standard anti-TB chemotherapy; following routine diagnosis, participants were transferred for RASC sampling, ethical approval having sanctioned a 2-hour delay to initiation of standard TB chemotherapy to enable bioaerosol collection.

## Sample collection

Bioaerosol collection was done as previously described [12] with the sampling method improved by using a Coriolis μ Biological Air Sampler (Bertin Technologies SAS, France) possessing the capacity to capture up to 500 L of expired air. This enabled the concentration of <500 L of expired air per subject per hour into ~5–10 mL sterile phosphate-buffered saline (PBS), thus ensuring high sampling volumes while producing a tractable, low-volume liquid output for downstream manipulation and analysis.

## Bacterial culture conditions and DMN-trehalose staining

The laboratory strain, *Mtb* H37RvMA [37], was grown at 37˚C in Middlebrook 7H9 (Difco) liquid broth supplemented with 0.2% (v/v) glycerol, 10% (v/v) Middlebrook OADC enrichment and 0.05% (w/v) Tween80 (Sigma-Aldrich). *Corynebacterium striatum* was cultured in LB broth (Sigma-Aldrich) at 37˚C.

The solvatochromic probe, 4-*N*,*N*-dimethylamino-1,8-napthalimide-trehalose (DMN-trehalose) [22], was used for all staining. Enzymatic incorporation of DMN-trehalose into the hydrophobic mycomembrane by the mycobacterial Antigen-85 complex enhances DMN fluorescence one thousand-fold, limiting background noise attributable to unincorporated probe and circumventing the need for multiple washes. For staining of exponentially replicating and stationary-phase bacilli, *Mtb* H37Rv cultures were grown to an $OD_{600}$ ~0.5 and ~1.2, respectively, before staining with 100 μM DMN-trehalose for 2 h. Thereafter, cells were harvested by centrifugation at $13000 \times g$ for 5 min before resuspending in PBS prior to visualization.

## Staining of bioaerosol samples

The 5–10 mL bioaerosol samples were concentrated by centrifugation at $3000 \times g$ for 10 min (Allegra X-15R, Beckman Coulter). The pellet was resuspended in 200 μl fresh Middlebrook 7H9 medium and stained overnight (12–16 h), following which the stained sample was concentrated at $13000 \times g$ for 5 min and resuspended in 20 μl sterile filtered PBS.

## Nanowell arraying

Stained bioaerosol samples were arrayed in a custom-designed nanowell device (Edge Embossing), the superstructure of which consisted of two rows of eight wells (16 total) overlaid on an embossed cyclic olefin copolymer (COC) film (Fig 1A and 1B). The COC film contained the 50 x 50 μm nanowells which were arrayed ~140 μm apart center-to-center (Fig 1C). Each microwell therefore comprised approximately 1600 nanowells. Prior to inoculation, the device was plasma coated (Novascan) to counteract hydrophobicity. Following DMN-trehalose staining, the concentrated aerosol sample was added to a single microwell. Samples were loaded and plates sealed using an adhesive film (ThermoFischer Scientific) before centrifuging at $3000 \times g$ for 10 min to disperse the sample for imaging.

## Serial imaging of putative *Mtb* in bioaerosols

As described above, the 5–10 mL bioaerosol samples were concentrated by centrifugation at $3000 \times g$ for 10 min (Allegra X-15R, Beckman Coulter). The pellet was resuspended in 200 μl fresh Middlebrook 7H9 medium and stained overnight (12–16 h), following which the stained sample was concentrated at $13000 \times g$ for 5 min and resuspended in 20 μl fresh Middlebrook 7H9 medium and incubated at 37 °C without shaking. Images were captured as described in fluorescent microscopy every 24 h for the first week (excluding weekends) and again on day 14 post isolation, with incubation at 37 °C throughout.

## Fluorescence microscopy

Imaging was done on a Zeiss Axio Observer 7 equipped with a 100× plan-apochromatic phase 3 oil immersion objective with a numerical aperture of 1·4. Epifluorescent illumination was provided by a 475 nm LED and non-specific fluorescence was removed with a Zeiss 38 HE filter set. Images were acquired using the Zeiss Zen software, and quantitative data extracted using MicrobeJ [38]. For serial imaging of bioaerosol samples, re-identification of putative bacilli detected at Day 0 was done by determining the x-y coordinates of the specific nanowell relative to the top-most, center nanowell in the macro well.

## Statistical analysis

Data were exported from MicrobeJ and analyses performed using R, version 3.5.1. Data normality was assessed visually and, where applicable, a Wilcoxon Rank-Sum test was performed.

## Supporting information

**S1 Fig. Exclusion of non-mycobacterial, DMN-tre[+] organisms detected in bioaerosol samples based on cell morphology and staining profile.** (A) Representative images of (i) *C. striatum* cultured in LB broth during log-phase and stained with 100 uM DMN-trehalose for 5 min, and (ii) DMN-tre[+] organisms detected within a bioaerosol sample. (B) A panel of non-*Mtb*, DMN-tre[+] organisms, as determined by our inclusion criteria, identified in various bioaerosol samples. Scale bar, 5 μm.
(TIF)

**S2 Fig. Debris commonly found within TB bioaerosols.** Representative images of the three major categories of debris found within bioaerosol samples after overnight staining with 100 μM DMN-trehalose and visualization within a 50 x 50 μm nanowell. (A) Large, crystalline debris, (B) small fluorescent debris, and (C) granular debris.
(TIF)

## Author Contributions

**Conceptualization:** Ryan Dinkele, Sophia Gessner, Carl Morrow, Gilla Kaplan, Valerie Mizrahi, Robin Wood, Digby F. Warner.

**Data curation:** Ryan Dinkele, Sophia Gessner, Andrea McKerry, Bryan Leonard, Ronnett Seldon, Carl Morrow.

**Formal analysis:** Ryan Dinkele, Sophia Gessner, Anastasia S. Koch, Carl Morrow, Robin Wood, Digby F. Warner.

**Funding acquisition:** Valerie Mizrahi, Robin Wood, Digby F. Warner.

**Investigation:** Ryan Dinkele, Sophia Gessner, Wayne Bryden, Charles Call, Robin Wood, Digby F. Warner.

**Methodology:** Ryan Dinkele, Sophia Gessner, Andrea McKerry, Bryan Leonard, Ronnett Seldon, Anastasia S. Koch, Brian Smith, Courtney McLoud, Andrew Kamholz, Wayne Bryden, Charles Call, Digby F. Warner.

**Project administration:** Ryan Dinkele, Sophia Gessner, Ronnett Seldon, Carl Morrow, Melitta Gqada.

**Resources:** Mireille Kamariza, Carolyn R. Bertozzi, Brian Smith, Courtney McLoud, Andrew Kamholz, Wayne Bryden, Charles Call.

**Supervision:** Valerie Mizrahi, Robin Wood, Digby F. Warner.

**Visualization:** Ryan Dinkele, Andrea McKerry, Bryan Leonard.

**Writing – original draft:** Ryan Dinkele, Sophia Gessner, Anastasia S. Koch, Digby F. Warner.

**Writing – review & editing:** Ryan Dinkele, Sophia Gessner, Anastasia S. Koch, Carl Morrow, Mireille Kamariza, Gilla Kaplan, Valerie Mizrahi, Robin Wood, Digby F. Warner.

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
