## [Decision Letter · Decision Letter 0]

19 Aug 2020

Dear Prof. Warner,

Thank you very much for submitting your manuscript "Capture and visualization of live Mycobacterium tuberculosis bacilli from tuberculosis bioaerosols" for consideration at PLOS Pathogens. As with all papers reviewed by the journal, your manuscript was reviewed by members of the editorial board and by several independent reviewers. In light of the reviews (below this email), we would like to invite the resubmission of a significantly-revised version that takes into account the reviewers' comments.

All reviewers agreed on the importance of this topic and expressed enthusiasm for the approach.  However, there was also general agreement that the manuscript could be improved.  In particular, the clinical characterization of the study population should be enhanced, and several claims either need to be tempered or supported with additional data.  

We cannot make any decision about publication until we have seen the revised manuscript and your response to the reviewers' comments. Your revised manuscript is also likely to be sent to reviewers for further evaluation.

Sincerely,

Christopher M. Sassetti

Associate Editor

PLOS Pathogens

Sabine Ehrt

Section Editor

PLOS Pathogens

Kasturi Haldar

Editor-in-Chief

PLOS Pathogens

orcid.org/0000-0001-5065-158X

Michael Malim

Editor-in-Chief

PLOS Pathogens

orcid.org/0000-0002-7699-2064

Reviewer's Responses to Questions

**Part I - Summary**

Reviewer #1: This paper presents very novel and potentially exciting methods and data, but unfortunately it has many weaknesses that the authors do not acknowledge. I strongly recommend the includsion of a 'limitations' section of paragraph towards the end. My major concern is the lack of any culture data on the patients samples, in spite of the title suggesting that there is capture of 'live M. tuberculosis.' Ideally it would have seemed reasonable to compare the bioaerosol collected to that collected by an Andersen cascade impactor, as used in other studies. But at minimum the collection from the Coriolis sampler could have been cultured. Why was this not done? The use of DMN trehalose has only been done on sputum samples and not previously on aerosols, which may contain bacilli that are more stressed and may not have the same responses as those in sputum.

In a similar vein, even though the RASC helps reduce mold contamination from the outside environment, cough aerosols have contained other organisms. In the original cough aerosol paper (Fennelly et al, 2004), nontuberculous mycobacteria, diptheroids, Cladosporium and other Gram-positive and Gram-negative pathogens were isolated from the patients. How can you confirm that the DMN trehalose stain is so specific that it is not staining some of the patients' other respiratory flora?

The study population is poorly characterized. Patients who are Xpert-positive are likely, but not always culture positive. Given the traditional use of AFB smears and cultures, these simple tests would have helped characterize the population better for readers. It is also usually customart to include some description of imaging, at least whether or not there was cavitary disease.

In line 202, reference 17 is not the best source for the statement, as that paper does not include anything about infectiousness or transmission. That statement can also be challenged, as one study found that cough aerosol cultures of Mtb were the best predictors of infectiousness and transmission of new infections to household contacts (Jones-Lopez et al, 2013). In addition, ref #17 is not the best choice in line 103, as the method used was originally published by Honeybone in the J Clin Microbiol in 2011.

Reviewer #2: The paper by Dinkele and colleagues details new methods for the detection of viable Mycobacterium tuberculosis in bioaerosols. The controlled nature of the air sampling procedure, coupled with a novel use of a fluorogenic substrate as a vital stain, makes this study extremely valuable to the field. The major finding, that the release of viable organisms, is significantly greater than were estimated by previous sampling methods, is important to the field. Moreover, this protocol has several potential applications linked to therapeutic interventions as a much more informative and sensitive “time to sputum negativity” type assay.

Reviewer #3: Dinkele, Gessler et al. report the development of a novel device (a nanowell arrayed microscope slide) which, when combined with staining by a solvatochromic dye (DMN-trehalose) allows quantitative visualization of exhaled Mtb from patient bioaerosols. The respiratory aerosol sampling chamber (RASC) has been previously described (PLoS One 2016) and the sensitive capture of Mtb (as identified by culture and/or PCR) from patient bioaerosols was reported in 2018 (Gates Open Research). The current publication introduces the nanowell capture device paired with the application of the DMN-trehalose dye and shares the morphologic and staining characteristics of the captured organisms. While the reviewer is in principle strongly supportive of aerobiology research which s(he) views as pioneering work on the frontiers of modern microbiology, the data included in the manuscript are descriptive, without definitive link to mechanism or novel insight into pathogenesis. Inclusion of downstream analysis (whole genome sequencing, proteomics, metabolomics, other), from even only a few patient aerosol samples would suffice to increase interest in publication to inform and inspire the broader microbiology community.

**Part II – Major Issues: Key Experiments Required for Acceptance**

Reviewer #1: My comments above might be interpreted to say that culture from the Coriolis should absolutely be done prior to acceptance. I suspect that may be impossible at this point. There is enough exciting potential in these methods that i would hate to see this not published at all. But the authors clearly need to describe the limitations in the paper and to explain some of the reasons for not doing some of what I suggested.

The sentence from line 105 to 108 is very confusing and needs re-writing for clarity. Although cough obviously assumes symptomatic spread, cough has been shown to be associated with 'natural transmission' by both R. Loudon and R. Turner.

In addition to addressing the limitations mentioned above, in your concluding paragraph of the discussion, it would be helpful to address how you plan to link these findings to transmission, e.g. in a household contact study?

Reviewer #2: The paper is written in a clear and well-balanced manner, the data are described with care and rigor, and the short-comings and caveats with respect to specificity and heterogeneity are discussed fully. One could ask for more data to be included but I feel this would not alter the basic findings or the immediate utility of this method. I believe it more important that these findings should be published quickly.

Reviewer #3: lines 34-35, although it is correct that ‘the contribution of asymptomatic transmitters to the TB pandemic is overlooked’ this is not addressed by the data presented in the manuscript and should be removed from the abstract. It however represents a valid and important area of future research that is appropriately included in the discussion.

Lines 43-44 ‘variations in DMN-trehalose incorporation suggested metabolic heterogeneity in aerosolized Mtb’ the heterogeneity depicted in Figure 4 is difficult for the reviewer to perceive. Can more distinct images be included? Can the heterogeneity be quantified via imaging software and statistics applied to confirm the presence of three distinct phenotypes? In addition, authors attribute heterogeneity in DMN-trehalose staining to metabolic state; inclusion of DCTB staining or staining of organisms undergoing in vitro antimycobacterial drug exposure for direct comparison would more strongly support this claim (Kamarizza Sci Trans Med 2018, Figure 7)

Lines 44-46 ‘intrapatient comparisons indicated that Mtb bioaerosols were probably derived from a compartment other than sputum’. This is conjecture and not well-supported by the data which show only a minor length difference between bioaerosol-derived and sputum-derived organisms. This difference could be due to sampling method as Coriolis subjects bioaerosols to manipulation that differs from standard sputum sampling. This claim requires additional data to support (whole genome sequencing, metabolomics, proteomics) or, further clarification of the basis upon which the claim is made.

Line 164, the major finding of the manuscript is that staining with DMN-Tre (89% positive) closely approximates the previously published culture/PCR positivity rates in exhaled bioaerosols from TB patients (93%, Gates Open Research 2018) while still enabling down-stream analysis. Some evidence of that downstream analysis (whole genome sequencing, metabolomics, proteomics or other) should be included in the manuscript as the reported findings otherwise represent an important but incremental advance beyond the prior publications describing the RASC (Plos One 2016) and the capture of viable Mtb from patient bioaerosols (Gates Open Research 2018) which in and of itself is not of broad interest to the wider microbiology community.

**Part III – Minor Issues: Editorial and Data Presentation Modifications**

Reviewer #1: Line 67, the ref #1 should be placed at the end of that sentence.

Line 108: there is no period at the end of the sentence.

Line 134: 'bespoke' is a rather obscure word. Why not say something more direct, e.g. 'custom'?

Reviewer #2: (No Response)

Reviewer #3: Might alter the title to “Capture and visualization of live Mycobacterium tuberculosis bacilli from patient bioaerosols”

Lines 51-53 ‘ongoing transmission…is the primary driver of incident disease’. Important to provide citation here as this may well be true in high transmission settings in Sub-Saharan Africa but is controversial in other geographies, for example China, where some models support reactivation as the driver of the epidemic.

Lines 53-57 include citation here as well. this paragraph reads very similarly to Andrews ERJ 2019 which shares co-authors with this manuscript. This may well be due to the nature of introductions about TB transmission but authors should take care not to repeat themselves too closely.

Line 65 interventions “to better understand and ultimately interrupt Mtb transmission”?

Line 146 please address the differences in width in the discussion section (what is the likely source and significance of this, if any)?

Figure 2C has both columns i and ii as well as two rows; the columns are labelled but the rows are not; how does the top row complement or differ from the bottom row?

Figure 2D is not described in the text; which non-Mtb, DMN-tre+ organisms are in which row and column? This should be clarified in the legend.

Line 162 please confirm that the 28 patients reported here are not a subset of the 35 patients previously presented in the 2018 Gates Open Research publication

Figure 3 image ii (patient) makes more sense first followed by image i (RASC).

Are there any differences in any parameter based on strain lineage?

Table 1, what is meant by NA in the table? Please add a footnote clarifying this.

HIV+/HIV- seem not to differ in Mtb counts; this is unexpected (Kwan and Ernst Clin Micro Rev 2011) and should be addressed in the discussion

Lines 168-173 as noted above (abstract) the three patterns are indistinguishable to the reviewer. Please provide higher resolution images and quantitative analysis; also please include an image of PBS-starved organisms for direct comparisons (currently the reader has to ‘take your word for it’).

Line 175 please include more detail in the discussion about the potential significance of patient-derived organisms being shorter than organisms in log phase growth

Lines 179-182 please include data demonstrating non-Mtb DMN-Tre+ organisms in a supplemental figure

Line 186 because no physical or biochemical analysis of particulate matter was performed, Figure 5 should be relegated to a supplemental figure and the text shortened accordingly; if, in contrast such analysis exists it could be included in the main text (identification of host-associated molecules, mucus or metabolites for example)

Lines 210-215 why are only 3 of the 4 patients shown? Very difficult to observe polar staining differences and therefore would benefit from inclusion of a summary figure as is included for length. Also the potential mechanisms underlying differences in growth/polar staining between cultured organisms and patient bioaerosol should be more clearly elucidated in the discussion.

Line 231 results section suggests that some of the particles arise from Tyvek suits; because analysis of non-microbial constituents is not reported it is not possible to determine whether these particles arose from patient breath or other non-respiratory compartment that is of little relevance to TB pathogenesis

Lines 240-246 do not make sense to the reviewer. How can length be directly linked to cavity surface as the source of the organism? This conclusion seems to require substantially more evidence than is included in the manuscript.

Lines 293-296 “closing the loop” is unclear. Could bioaerosols be used to infect guinea pigs as a measure of infectiousness? Reference could be made to aerobiology studies from the 1890s (comprehensively reviewed by Peter Donald in an NIH webinar http://www.stoptb.org/assets/documents/news/TCRB%20Presents.pdf and manuscript published in IJTLD 2018). The last sentence could reference multiomic approaches to be applied to viable organisms isolated on a single cell basis directly from patient aerosols (proteomics, metabolomics, whole genome sequencing) which represent the frontier of modern microbiology.

References

Line 369, too many spaces between commas

Line 419, immunopathology in ?

Line 380, Articles?

Figure 3, vii the include/exclude distinction in the figure is uninterpretable, please include clearer examples and clarify rationale in the text (if software based, how was the software trained?)

Figure 4, what proportion of patients exhaled clumps? Was there a correlation with any clinical parameter? What proportion of total organism burden per patient with clumps was exhaled in clumps versus single organisms? What is the likely source and significance of this? What are the implications for measurement of drug response? Figure 4d what is the significance of cultured organisms being longer than aerosol? Why do you think this is the case?

PLOS authors have the option to publish the peer review history of their article (what does this mean?). If published, this will include your full peer review and any attached files.

Reviewer #1: **Yes: **Kevin P Fennelly

Reviewer #2: No

Reviewer #3: No
---

## [Decision Letter · Decision Letter 1]

28 Dec 2020

Dear Prof. Warner,

We are pleased to inform you that your manuscript 'Capture and visualization of live Mycobacterium tuberculosis bacilli from tuberculosis patient bioaerosols' has been provisionally accepted for publication in PLOS Pathogens.

Best regards,

Christopher M. Sassetti

Associate Editor

PLOS Pathogens

Sabine Ehrt

Section Editor

PLOS Pathogens

Kasturi Haldar

Editor-in-Chief

PLOS Pathogens

orcid.org/0000-0001-5065-158X

Michael Malim

Editor-in-Chief

PLOS Pathogens

orcid.org/0000-0002-7699-2064

Reviewer Comments (if any, and for reference):

Reviewer's Responses to Questions

**Part I - Summary**

Reviewer #1: The revised manuscript is markedly improved over the original.

Reviewer #4: The reviewer appreciates the authors' thoughtful response to review as evidenced in the revised manuscript, figures and cover letter. In particular, updated Figures 2 and 4 as well as the addition of Figure 5 demonstrate the potential of the RASC+nanowell system to make novel measurements that may lead to breakthrough insights in microbiology.

**Part II – Major Issues: Key Experiments Required for Acceptance**

Reviewer #1: None.

Reviewer #4: All issues adequately addressed in the response to review.

**Part III – Minor Issues: Editorial and Data Presentation Modifications**

Reviewer #1: No further suggestions.

Reviewer #4: These have been addressed by the authors.

PLOS authors have the option to publish the peer review history of their article (what does this mean?). If published, this will include your full peer review and any attached files.

Reviewer #1: **Yes: **Kevin P. Fennelly

Reviewer #4: No

---

## [Editor Report · Acceptance letter]

26 Jan 2021

Dear Prof. Warner,

We are delighted to inform you that your manuscript, "Capture and visualization of live Mycobacterium tuberculosis bacilli from tuberculosis patient bioaerosols," has been formally accepted for publication in PLOS Pathogens.

Best regards,

Kasturi Haldar

Editor-in-Chief

PLOS Pathogens

orcid.org/0000-0001-5065-158X

Michael Malim

Editor-in-Chief

PLOS Pathogens

orcid.org/0000-0002-7699-2064